# Genetic mapping and evolutionary analysis of human-expanded cognitive networks

Yongbin Wei[1], Siemon C. de Lange [1], Lianne H. Scholtens [1], Kyoko Watanabe [2], Dirk Jan Ardesch [1], Philip R. Jansen [2,3], Jeanne E. Savage [2], Longchuan Li[4], Todd M. Preuss[5,6,7], James K. Rilling[6,7,8,9,10], Danielle Posthuma[2,11] & Martijn P. van den Heuvel [1,11]*

Cognitive brain networks such as the default-mode network (DMN), frontoparietal network, and salience network, are key functional networks of the human brain. Here we show that the rapid evolutionary cortical expansion of cognitive networks in the human brain, and most pronounced the DMN, runs parallel with high expression of human-accelerated genes (HAR genes). Using comparative transcriptomics analysis, we present that HAR genes are differentially more expressed in higher-order cognitive networks in humans compared to chimpanzees and macaques and that genes with high expression in the DMN are involved in synapse and dendrite formation. Moreover, HAR and DMN genes show significant associations with individual variations in DMN functional activity, intelligence, sociability, and mental conditions such as schizophrenia and autism. Our results suggest that the expansion of higher-order functional networks subserving increasing cognitive properties has been an important locus of genetic changes in recent human brain evolution.

[1] Connectome Lab, Department of Complex Trait Genetics, Center for Neurogenomics and Cognitive Research, Amsterdam Neuroscience, Vrije Universiteit Amsterdam, 1081 HV Amsterdam, The Netherlands. [2] Department of Complex Trait Genetics, Center for Neurogenomics and Cognitive Research, Amsterdam Neuroscience, Vrije Universiteit Amsterdam, 1081 HV Amsterdam, The Netherlands. [3] Department of Child and Adolescent Psychiatry, Erasmus Medical Center, 3015 GD Rotterdam, The Netherlands. [4] Marcus Autism Center, Children's Healthcare of Atlanta, Emory University School of Medicine, Atlanta, GA 30322, USA. [5] Division of Neuropharmacology and Neurologic Diseases, Emory University, Atlanta, GA 30322, USA. [6] Center for Translational Social Neuroscience, Emory University, Atlanta, GA 30322, USA. [7] Yerkes National Primate Research Center, Emory University, Atlanta, GA 30322, USA. [8] Department of Anthropology, Emory University, Atlanta, GA 30322, USA. [9] Center for Behavioral Neuroscience, Emory University, Atlanta, GA 30322, USA. [10] Department of Psychiatry and Behavioral Sciences, Emory University, Atlanta, GA 30322, USA. [11] Department of Clinical Genetics, Amsterdam Neuroscience, Amsterdam UMC, 1081 HV Amsterdam, The Netherlands. *email: martijn.vanden.heuvel@vu.nl

The human brain is capable of supporting a wide range of complex cognitive abilities, more so than other highly developed and intelligent great apes, such as the chimpanzee, one of our closest living evolutionary relatives with which we share the majority of our genetic material[1]. This distinction in cognitive abilities is commonly believed to be associated with the rapid expansion of multimodal association areas and their structural and functional connections in the human brain[2–4], with cognitive functional networks, such as the frontoparietal network, salience network, and default-mode network (DMN), playing an essential role in higher-order brain functions[5–7]. These cognitive functional networks are highly heritable[8,9] and relate to genetic effects associated with neuron growth and metabolism[10]. Uncovering the evolutionary genetic underpinnings of cognitive functional networks, and in particular, to what extent cognitive functional networks have developed in recent human evolution, is crucial for our understanding of the high cognitive complexity of human brain function.

The DMN in particular has been identified as a central network in human cognition, consisting of densely connected areas such as the posterior cingulate, precuneus, inferior parietal, middle temporal, and medial prefrontal cortices[5,6]. Comparative neuroimaging studies have shown default-mode activity in chimpanzees[11] and macaques[12], but with potential subtle differences in both the spatial and topological layout of this central network[13]–changes that may relate to enhancement of cognitive functions in humans compared to other primate species. The DMN is central to social cognition, including aspects of mental self-projection[14], mental rehearsal of future actions[15], and understanding of another person's mental perspective. These advanced social abilities are likely to have been highly adaptive during recent human evolution[16], potentially enabling humans to make more complex social inferences[14].

Here, we studied the expansion and evolutionary genetics of higher-order cognitive networks in recent human brain evolution, with a particular focus on the evolutionarily genetic drive of the DMN. Recent genome-wide studies comparing the human genome with that of the chimpanzee have identified a unique set of loci that displayed accelerated divergence in the human lineage[17,18]. Genes associated with these so-called human accelerated regions (HAR) have been linked to neuron development, but also the development of brain disorders such as autism spectrum disorder (ASD)[19]. We integrate genetic data with comparative neuroimaging and present that regions of higher-order cognitive networks are highly expanded in recent human brain evolution. We show that HAR genes likely have played a crucial role in this, being highly expressed in expanded cognitive networks (and in particular the DMN) and being differentially expressed in the human brain compared to chimpanzees and macaques. We provide further evidence of HAR and DMN genes to be important in human cognitive functioning, social behavior, and mental disorders, such as autism and schizophrenia.

## Results

**Human cortical expansion**. We started by mapping the expansion of the human cortex (*Homo sapiens*) compared to the cortex of the chimpanzee (*Pan troglodytes*), one of our closest living evolutionary relatives along with the bonobo (*Pan paniscus*). Cortical morphometry of the chimpanzee and human cortex was assessed using a surface-to-surface mapping of 3D reconstructions of the cortical mantle across both species, based on in vivo T1-weighted MRI (29 chimpanzees, 30 humans; Fig. 1a). The largest expansion of the human cortex was found in areas of bilateral orbital inferior frontal gyrus (×4.0 expansion), rostral middle frontal lobe (×3.8 expansion), inferior/middle temporal

lobe (×3.0/2.9 expansion), lingual gyrus (×2.9 expansion), right inferior parietal lobe (×3.7 expansion), and left precuneus (×2.7 expansion; two-sample $t$-test on the normalized expansion, $q < 0.001$, false discovery rate [FDR] corrected; Cohens'$d > 0.989$; Fig. 2a). The lowest expansion was found in primary areas, including bilateral precentral gyrus (×1.3 expansion), postcentral gyrus (×1.4 expansion), and paracentral lobe (×1.2 expansion; $q < 0.001$, FDR corrected; Cohens'$d < -1.047$; Fig. 2a and Supplementary Data 1).

We next grouped cortical areas into the visual (VN), somatomotor (SMN), dorsal-attention (DAN), limbic (LN), ventral-attention (VAN, also commonly referred to as the salience network), frontoparietal (FPN), and default-mode network (DMN) (Fig. 2b and Supplementary Methods)[20]. Higher-order cognitive networks (i.e., DMN, FPN, VAN) displayed particularly high levels of cortical expansion as compared to the SMN/VN (×1.2 larger expansion in regions of higher-order cognitive networks combined compared to the regions of the SMN/VN combined, two-sample $t$-test, $t(86) = 3.257$, $p = 0.002$; Fig. 2d). FPN showed the largest expansion (mean: ×2.9 expansion), with the DMN in second place (mean: ×2.4 expansion), showing both significantly higher expansion when comparing each of them with the rest of the brain (FPN: $t(108) = 3.360$, $p = 0.001$; DMN: $t(108) = 2.621$, $p = 0.010$; FDR corrected; Supplementary Table 1). In contrast, separately examining the other five networks did not show significant increases in the expansion of these networks compared to the rest of the cortex.

**HAR gene expression**. We then examined this distinct pattern of human cortical expansion across the seven resting-state functional networks in relation to cortical gene expression patterns relevant to human evolution. Microarray data on gene expression across cortical regions were obtained from the Allen Human Brain Atlas (AHBA) (http://human.brain-map.org/), containing transcriptional profiles of 20,734 genes across 57 areas of the left cortical mantle (Fig. 1c). Genes relevant to human evolution were taken as the list of 2143 genes associated with HAR as presented previously by Doan and colleagues[19], selected based on positional mapping. Alternative selection and allocation of HAR-associated genes are possible (e.g., using chromatin interactions) and we examined such alternatives to validate our results (Supplementary Note 1).

Transcription data of 1711 HAR-associated genes were present in AHBA (referred to as HAR genes; Supplementary Data 2), and we further examined their cortical gene expression levels in comparison to the cortical expansion. The expression profile of HAR genes was positively correlated to the pattern of human cortical expansion (Pearson's $r(53) = 0.360$, $p = 0.007$; Supplementary Fig. 1a), indicating the highest HAR gene expression in highly expanded areas of the human cortex. This association significantly exceeded the null condition of correlations between cortical expansion and expression of random gene sets (i.e., 1711 random genes) selected from a pool of 8686 genes related to general evolutionarily conserved genetic elements (ECE genes;[21] $p < 0.001$, permutation test, 10,000 permutations; Supplementary Fig. 1a). Cortical regions of cognitive networks also showed significantly higher expression of HAR genes compared to regions of the SMN/VN ($t(44) = 2.742$, $p = 0.009$; FDR corrected; Supplementary Fig. 1b), with regions of the DMN showing the highest HAR gene expression ($t(55) = 2.274$, $p = 0.027$, uncorrected; comparing the DMN to all other networks combined; Supplementary Fig. 1c). These effects again significantly exceeded the null conditions of effects of random ECE genes (all $p < 0.001$, 10,000 permutations). Furthermore, examining the other six

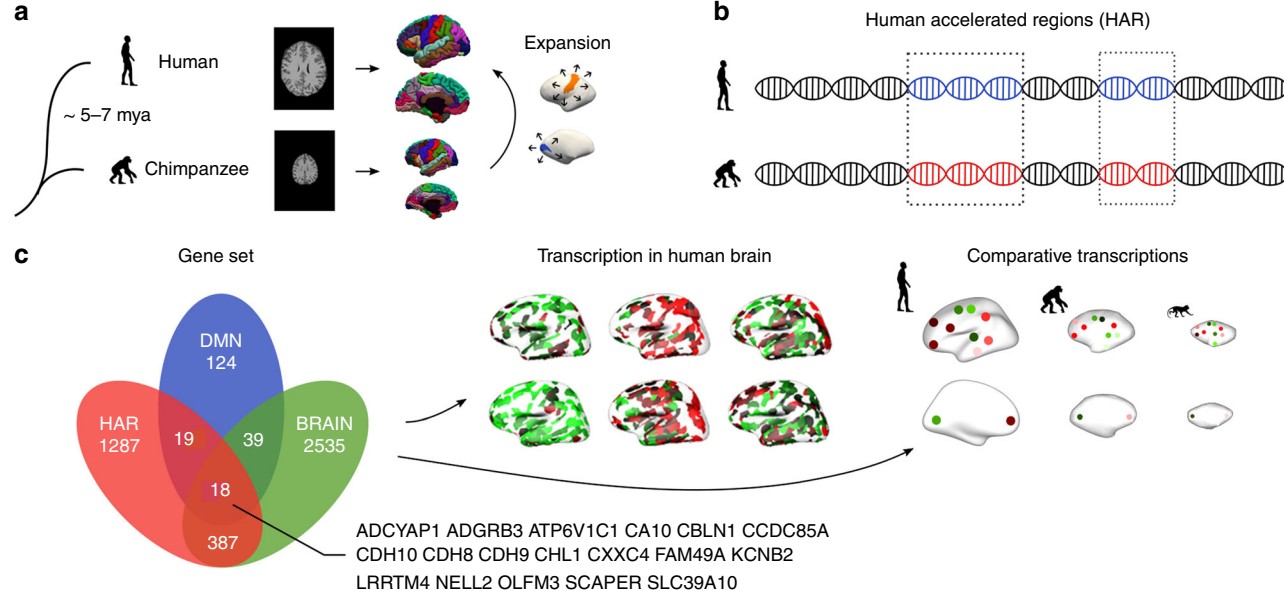

**Fig. 1** Methods overview. **a** The human and chimpanzee cortex were constructed using MRI data, with chimpanzee-to-human cortical expansion computed based on the reconstructed cortical maps. **b** Genes associated with human accelerated regions (HAR), which represent genomic loci with accelerated divergence in humans, were examined. **c** Cortical gene expression of HAR genes and HAR-BRAIN genes were examined using human transcription data from the Allen Human Brain Atlas (AHBA) and comparative transcription data of the human, chimpanzee, and macaque from the PsychENCODE database

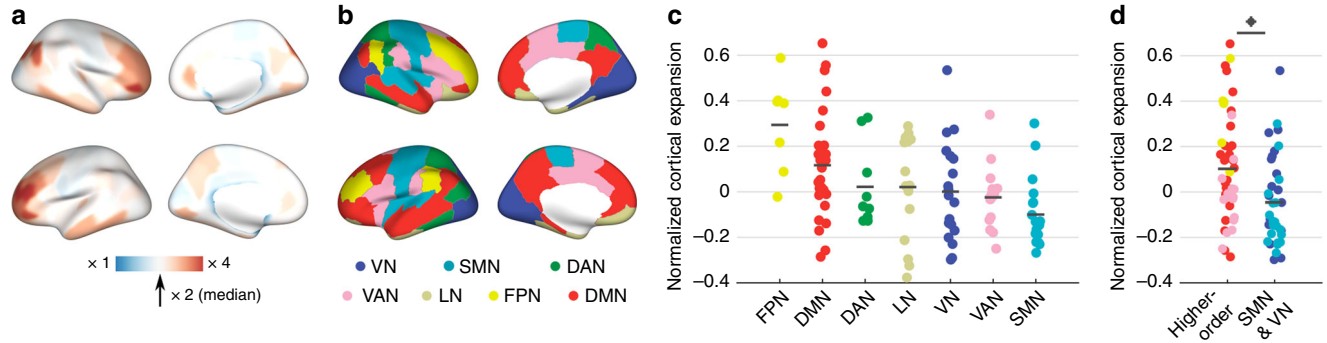

**Fig. 2** Cortical expansion. **a** Cortical expansion from chimpanzees to humans. Blue: below-median human expansion (i.e., < ×2 expansion compared to chimpanzee); red: above-median expansion (i.e., > ×2). **b** Brain maps of the seven resting-state functional networks according to the DK-114 atlas, describing the visual (VN), somatomotor (SMN), dorsal-attention (DAN), limbic (LN), ventral-attention (VAN), frontal-parietal (FPN), and default-mode network (DMN). **c** Levels of normalized cortical expansion per functional network in descending order of mean expansion. **d** Levels of normalized cortical expansion in higher-order cognitive networks (DMN, FPN, VAN) versus the SMN and VN. Dots depict cortical regions. Colors represent functional networks, as in panel **b**. Central marks are mean expansions. * indicates a two-sided *p*-value < 0.05, FDR corrected, two-sample *t*-test. Source data provided as Source Data file

functional networks separately did not show a significant enhancement of HAR gene expression (Supplementary Table 2).

**HAR-BRAIN gene expression.** With the set of HAR genes describing genes involved in all sorts of functions across the entire human body (and thus not specific to 'brain'), we continued by examining whether HAR genes related to brain processes may have played a specific role in the large cortical expansion of cognitive functional networks in human evolution. We identified genes commonly expressed in brain areas using the GTEx database (https://www.gtexportal.org/), selecting 2979 genes significantly more expressed in brain tissues compared to other available body sites (q < 0.05, FDR corrected; one-sided two-sample *t*-test; referred to as BRAIN genes); 415 genes (24.3%) out of the full set of 1711 HAR genes were observed to be significantly more expressed in brain tissues, a set from now on referred to as

HAR-BRAIN genes (in contrast to HAR-nonBRAIN genes; Supplementary Data 2).

We then aimed to examine (1) whether HAR-BRAIN genes were more expressed particularly in regions of higher-order cognitive networks compared to the total set of HAR genes, and (2) to what extent HAR-BRAIN genes were more expressed in regions of higher-order cognitive networks, more than an average set of genes related to general brain processes (i.e., BRAIN genes). First, the cortical expression pattern of HAR-BRAIN genes was significantly correlated with the pattern of human cortical expansion (*r*(53) = 0.488, *p* < 0.001; Fig. 3c). Furthermore, HAR-BRAIN genes showed significantly higher expression in regions of cognitive networks as compared to the SMN/VN (*t*(44) = 5.136, *p* < 0.001, FDR corrected; Fig. 3e), with the highest expression levels again observed in the DMN (*t*(55) = 3.267, *p* = 0.002, FDR corrected, DMN versus the rest of the cortex; Fig. 3f). These effects were, respectively, ×3.2 and ×2.7 larger than the

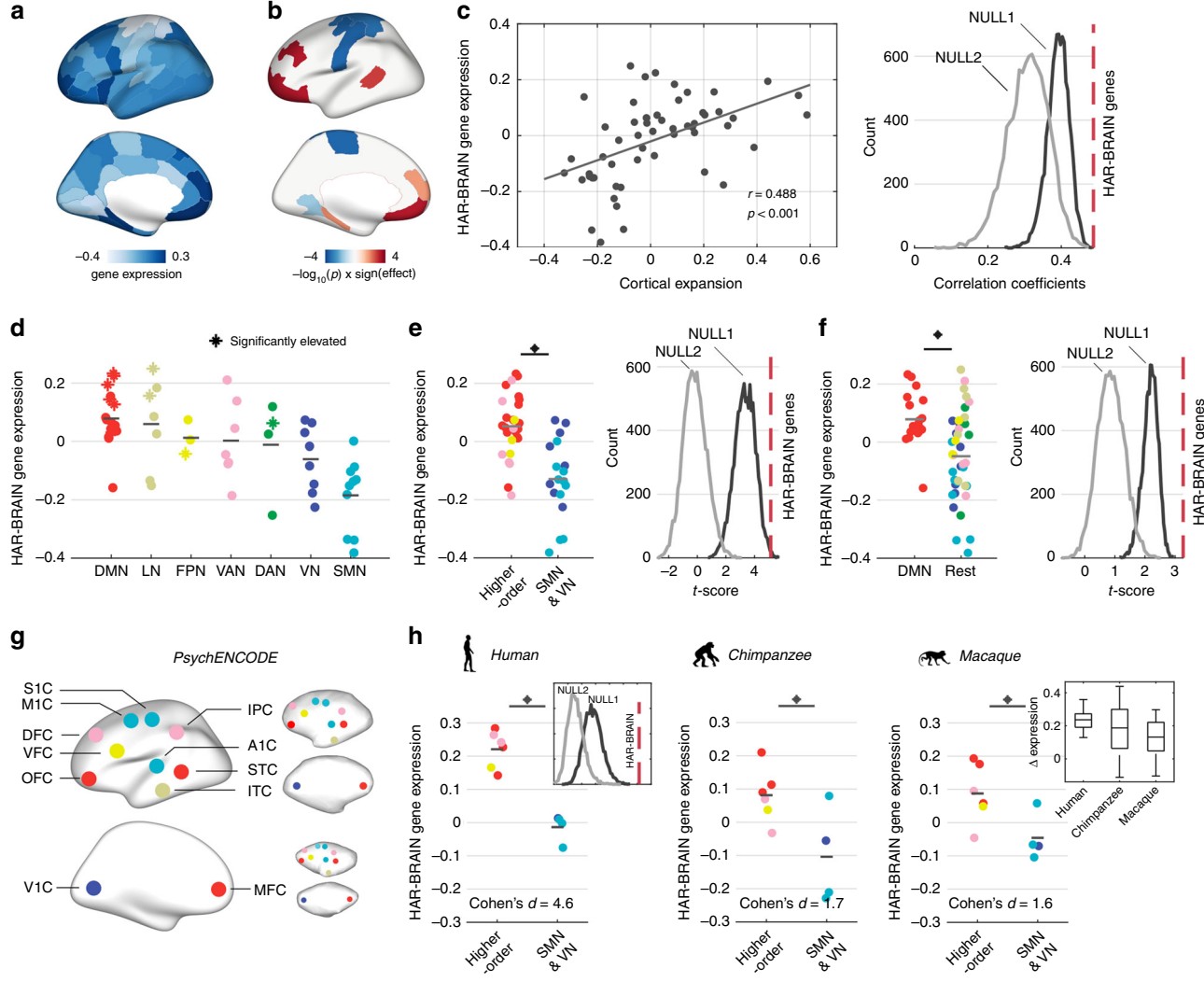

**Fig. 3** HAR-BRAIN gene expression. **a** Cortical gene expression of HAR-BRAIN genes. **b** Cortical maps of the significance level obtained by permutation tests, comparing expressions of HAR-BRAIN genes to equally sized random gene-sets taken from BRAIN (NULL1) and ECE genes (NULL2). **c** Association between the gene expression profile of HAR-BRAIN genes and normalized cortical expansion between human and chimpanzee (left). The correlation coefficient is significantly higher than NULL1 and NULL2 (both $p < 0.001$, two-sided, permutation test; right). **d** HAR-BRAIN gene expression within each of the seven functional networks ranked in descending order of the mean expression. Asterisks (*) indicates significantly upregulated regions as in panel **b**. **e** HAR-BRAIN gene expression in cognitive networks (DMN, FPN, and VAN) versus the SMN and VN (left), with permutation results demonstrated in the right panel (two-sided $p = 0.003$ and $p < 0.001$ for NULL1 and NULL2, respectively). **f** HAR-BRAIN gene expression in the DMN versus the rest of the cortex (left), with permutation results demonstrated in the right panel (two-sided $p < 0.001$ for both NULL1 and NULL2). **g** Species-homologous brain areas as presented in the PsychENCODE dataset for the human (left), chimpanzee (upper right), and macaque (lower right). **h** Normalized expression levels of HAR-BRAIN genes in regions of higher-order networks compared to areas of the SMN/VN in humans ($p < 0.001$, two-sample $t$ test). Largest differences in gene expression are found in humans, with chimpanzees in second place, followed by macaques ($p = 0.002$, Jonckheere-Terpstra test). Asterisks (*) indicates two-sided $p < 0.05$, FDR corrected. Central marks denote the mean gene expression. Boxplot center, median; box = 1st–3rd quartiles (Q); lower whisker, Q1–1.5 × interquartile range (IQR); upper whisker, Q3 + 1.5 × IQR. Colors indicate the assignment of functional networks, as in Fig. 2b. M1C primary motor cortex, S1C primary sensory cortex, IPC inferior parietal cortex, STC superior temporal cortex, ITC inferior temporal cortex, A1C primary auditory cortex, OFC orbital frontal cortex, VFC ventral frontal cortex, DFC dorsal frontal cortex, V1C primary visual cortex, MFC medial frontal cortex. Source data provided as Source Data file

effect obtained by HAR-nonBRAIN genes ($t(55) = 1.028$, $p = 0.309$ and $t(55) = 1.212$, $p = 0.232$, separately, Supplementary Fig. 2). Notably, examining the other six functional networks separately did not show any significant elevations of HAR-BRAIN gene expression (Supplementary Table 3), suggesting the highest expression level in the DMN.

Second, we compared HAR-BRAIN gene expression with two types of null-distributions of expression differences generated by randomly selecting the same number of genes (i.e., 415) from the pool of 2979 BRAIN genes (referred to as NULL1) and 8686 ECE

genes (referred to as NULL2). The elevated expression of HAR-BRAIN genes in regions of higher-order cognitive networks was significantly larger than both null distributions ($p = 0.006$ and $p < 0.001$ for NULL1 and NULL2, respectively; 10,000 permutations; Fig. 3e). The same result was observed when examining DMN regions specifically ($p < 0.001$ for both NULL1 and NULL2; 10,000 permutations; Fig. 3f), suggesting a specific role of HAR-BRAIN genes in differentiating DMN regions from the rest of the brain. Permutation testing based on randomly shuffling cortical areas showed similar results (Supplementary Fig. 3). An

exploratory examination on gene expression in each individual further demonstrated that the differentiation of HAR-BRAIN gene expression between the DMN and the rest of the cortex significantly correlated with the ratio of brain volume of the DMN regions ($r(4) = 0.839$, $p = 0.037$; Supplementary Fig. 4).

We then examined HAR-BRAIN expression for each of the cortical regions of the 7 functional networks separately; 10 cortical areas showed significantly high expression of HAR-BRAIN genes compared to a random selection of genes out of both BRAIN (NULL1) and ECE genes (NULL2) (FDR corrected, $q < 0.05$; 10,000 permutations; Fig. 3b). Importantly, 7 out of these 10 regions described regions of the higher cognitive networks and 6 out of these 7 regions described DMN regions ($p = 0.034$ and 0.008, respectively; hypergeometric test). These findings together suggest a specific role of HAR-BRAIN genes in the architecture of cognitive functional brain networks, beyond effects of general evolutionary conserved genes and general BRAIN genes.

**Chimpanzee-human comparative gene expression.** Our analyses so far suggested that HAR-BRAIN genes were more expressed in highly expanded regions of higher-order cognitive networks, but did not yet provide direct information on whether HAR-BRAIN gene expression is upregulated in the human brain compared to that of other primate species. To examine this, we used gene expression data from the PsychENCODE database (http://evolution.psychencode.org/)[22], which describes gene expression of 11 comparable cortical regions across the human, chimpanzee, and macaque. Due to the lower spatial sampling of cortical regions (data of 3 DMN regions available, Fig. 3g), we limited our examination to a comparison between cognitive networks and the SMN/VN. First, we replicated the observation of high expression of HAR-BRAIN genes in regions of higher-order cognitive networks compared to regions of the SMN/VN in humans ($t(8) = 7.135$, $p < 0.001$; Fig. 3h), with this effect significantly exceeding both NULL1 and NULL2 (both $p < 0.001$, 10,000 permutations). This confirmed our AHBA-based findings of high HAR-BRAIN gene expression in cognitive networks in the human brain. Second, the differentiating level of HAR-BRAIN gene expression between cognitive and primary areas observed in humans (Cohen's $d = 4.605$) was found to be ×2.7 larger than the effects found in chimpanzees (Cohen's $d = 1.695$) and ×2.8 larger compared to macaques (Cohen's $d = 1.616$). Chimpanzees and macaques showed only marginally higher HAR-BRAIN gene expression in regions of higher-order cognitive networks as compared to the SMN/VN (chimpanzees: $t(8) = 2.626$, $p = 0.030$; macaques: $t(8) = 2.504$, $p = 0.037$). Furthermore, the difference in effect size between humans and chimpanzees was larger than expected based on NULL1 and NULL2 (both $p < 0.001$; human-macaque: NULL1, $p = 0.026$ and NULL 2, $p = 0.090$, only trend-level, not significant [n.s.]; 10,000 permutations; Supplementary Fig. 5).

Further evaluation of this cross-species effect showed a significantly decreasing step-wise relationship of differences in HAR-BRAIN gene expression between regions of higher-order networks and the SMN/VN from humans (highest) to chimpanzees and macaques (lowest differentiating expression, Jonckheere–Terpstra test, $p = 0.002$). To reduce the influence of a relatively large variance of expression levels within chimpanzees and macaques (Fig. 3h), we performed a leave-one-out analysis (iteratively leaving out one region at a time) and confirmed a larger mean gene expression difference between cognitive network regions and primary regions in humans in comparison to chimpanzees and macaques (Supplementary Fig. 6). These findings thus suggest that humans display upregulated expression

of HAR-BRAIN genes in brain areas involved in cognitive brain function as compared to other primates.

**Top strongest differentiating DMN genes.** We continued by investigating the biological properties of genes showing the highest levels of expression in DMN regions out of all genes. For each gene in AHBA, we computed the upregulated level of gene expression in regions of the DMN by calculating the $t$-score for expressions of the selected gene in regions of the DMN against the rest of the brain. The top 200 highly expressed genes (i.e., genes showing the highest positive $t$-scores, referred to as DMN genes; all $p < 0.004$) were taken as the DMN's most differentiating genes (Supplementary Data 3). Out of the top 200 DMN genes, we identified 37 to be HAR genes, particularly including 18 HAR-BRAIN genes, which greatly exceeded the chance level of randomly selecting 37 or 18 out of 20,734 genes (both $p < 0.001$, hypergeometric test). We also examined the top 53 genes with $p < 0.0014$ (partial Bonferroni corrected, see Supplementary Methods) and top 469 genes with $p < 0.01$ (uncorrected), which revealed comparable findings (Supplementary Note 2).

To investigate whether the observed effect was restricted to the DMN, an additional permutation analysis was performed by shuffling region labels and re-computing the top genes for each of these random network assignments. We found the ratio of HAR genes in the set of top DMN genes to be significantly higher than the null condition ($p < 0.001$, 10,000 permutations), which confirmed a dominant role of HAR genes in DMN organization. To further examine potential DMN specificity, we also selected the top 200 genes showing the largest differentiating gene expression in each of the other functional networks compared to the SMN/VN. In contrast to the DMN (revealing 18 overlapping genes with the set of HAR-BRAIN genes), the top 200 gene sets identified by the VAN, DAN, FPN, and LN, comprised, respectively, only 12,8,7 and 7 HAR-BRAIN genes.

Gene-set enrichment analysis on the set of DMN genes using hypergeometric testing in the web-based platform FUMA[23] showed significant over-representation of DMN genes in Gene Ontology (GO) terms related to cellular components of dendrite ($p = 2.60 \times 10^{-5}$), somatodendritic compartment ($p = 4.49 \times 10^{-5}$), synapse ($p = 2.15 \times 10^{-4}$), perikaryon ($p = 7.45 \times 10^{-5}$) and neuron projection terminus ($p = 2.26 \times 10^{-4}$), as well as molecular functions of neuropeptide hormone activity and calcium activated potassium/cation channel activity ($p = 1.66 \times 10^{-5}$ and $1.32 \times 10^{-4}$; FDR corrected; Supplementary Table 4).

**GWAS on DMN functional activity.** We then wanted to examine whether HAR/HAR-BRAIN genes played a role in inter-subject variation in default-mode functional activity in today's human population. We performed a GWAS on 6,899 participants from the UK Biobank[24] (see Supplementary Methods) with the amplitude of fMRI time series of the independent component analysis (ICA)-based resting-state networks ("NET-MAT amplitudes 25"[25] as described in https://www.fmrib.ox.ac.uk/ukbiobank/) as the phenotypes of interest. Particularly, we focused on the amplitude of the ICA component #1 that resembles the DMN (referred to as DMN amplitude, Fig. 4a). GWAS results for all single-nucleotide polymorphisms (SNP) with minor allele frequency (MAF) > 0.005 were assessed (Fig. 4b). The quantile-quantile plot showed a linkage disequilibrium score regression [LDSC] intercept of 0.999 (standard error [s.e.] = 0.006), with an inflation level of $\lambda_{GC} = 1.005$ and mean $\chi^2$ statistic = 1.012. LDSC-based SNP heritability [$h^2_{SNP}$] was 0.09 [s.e. = 0.06]. We observed 3 independent ($r^2 < 0.1$) genome-wide significant SNPs ($p < 5 \times 10^{-8}$; using linear regression model; Fig. 4b) across 2 genomic loci (Fig. 4c).

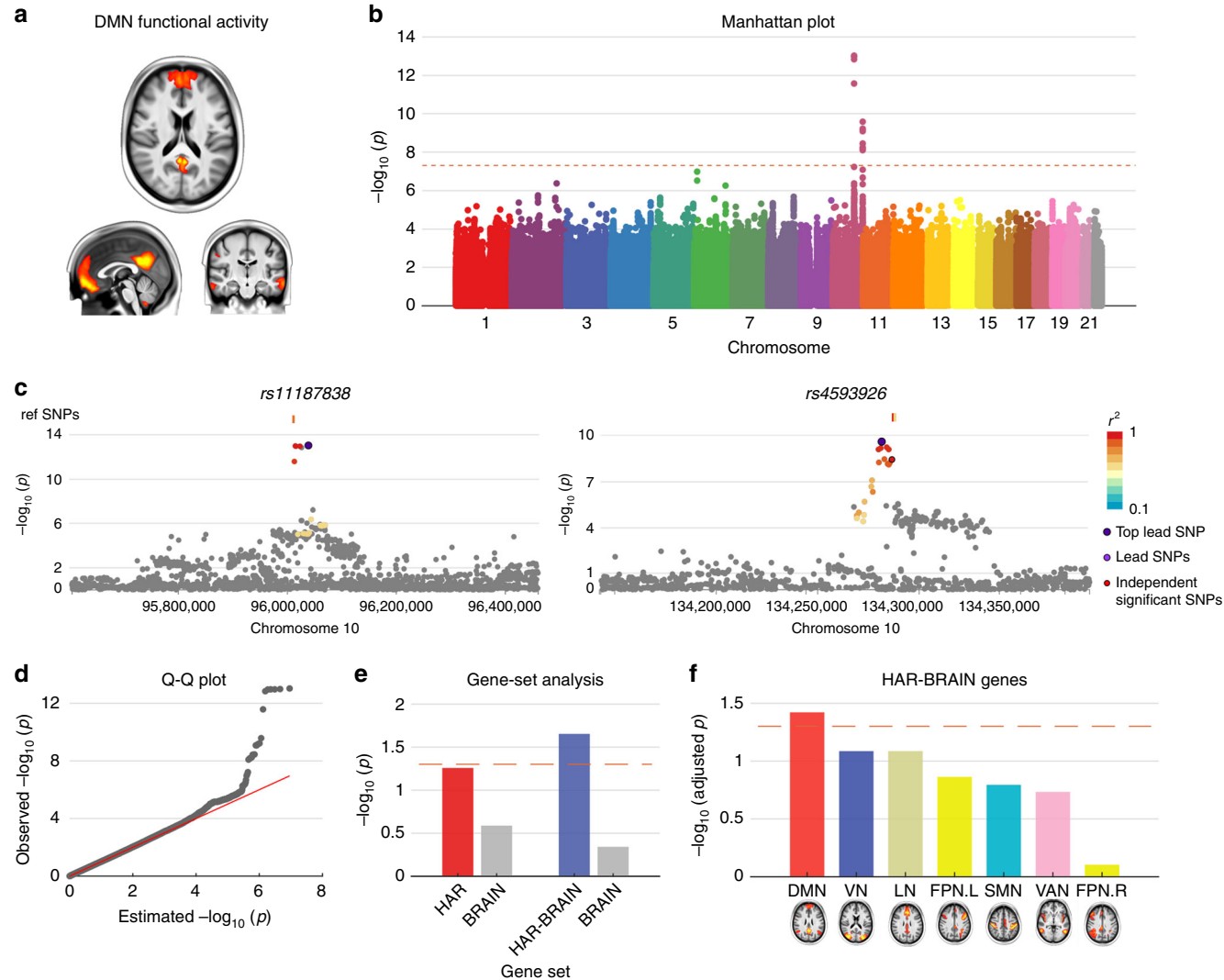

**Fig. 4** GWAS on DMN activity. **a** DMN component. **b** GWAS Manhattan plot showing $-\log_{10}$-transformed two-tailed $p$-value for all SNP ($y$-axis) and base-pair positions along the chromosomes ($x$-axis). Dotted red line indicates Bonferroni-corrected genome-wide significance ($p$-value $< 5 \times 10^{-8}$). **c** Regional plots of the two genomic loci (left, lead SNP: *rs11187838* and right, lead SNP: *rs4593926*). **d** Q–Q plot of SNP-based $p$-value in panel **b**. Observed $-\log_{10}$ transformed two-tailed $p$-values of associations with DMN functional activity are plotted against expected null $p$-values for all SNPs in the GWAS. **e** MAGMA conditional gene-set analysis. $-\log_{10}$-transformed $p$-values of the associations between HAR/HAR-BRAIN genes and DMN functional activity conditional upon BRAIN genes. Dashed line indicates $p = 0.05$. **f** MAGMA gene-set analysis on HAR-BRAIN genes and other "NETMAT amplitude 25" phenotypes representing functional activity in the other functional networks ($-\log_{10}$-transformed adjusted $p$-values, FDR corrected). Colors indicate the assignment of functional networks, as in Fig. 2b. Dashed line indicates adjusted $p = 0.05$

Furthermore, we annotated 19 significant SNPs ($p < 5 \times 10^{-8}$) with high LD ($r^2 \geq 0.6$) to the 3 independent SNPs using gene-mapping functions in FUMA[23] (see Methods section), which resulted in a set of 12 genes (Supplementary Table 5; three genes [*PLCE1, NOC3L, and SLC35G1*] annotated using brain-related eQTL and Hi-C mappings). Hypergeometric testing[23] showed significant enrichment of the 12 genes in the GWAS catalog[26] reported gene-set "plasma clozapine-norclozapine ratio in treatment-resistant schizophrenia" ($p = 1.26 \times 10^{-12}$; Supplementary Table 6). None of the genes overlapped with HAR-BRAIN genes or the top 200 DMN. One gene (*INPP5A*) denoted as a HAR gene.

We further investigated the potential association of HAR/HAR-BRAIN genes with variations in DMN amplitude using MAGMA linear-regression-based gene-set analysis[27]. We found HAR-BRAIN genes to be significantly associated with the phenotypic variation in DMN amplitude ($\beta = 0.015$, $p = 0.016$, FDR corrected). No significant effect was found for the set of HAR genes ($\beta = 0.011$, $p = 0.051$; Supplementary Table 7) or DMN genes ($\beta = 0.005$, $p = 0.219$). An additional conditional gene-set analysis[28] including the set of BRAIN genes as a covariate, further showed a significant association of HAR-BRAIN genes with variations in DMN amplitude ($\beta = 0.014$, $p = 0.022$; HAR genes: $\beta = 0.011$, $p = 0.055$; Fig. 4e). Furthermore, no significant effect was observed when we examined the association between HAR-BRAIN genes and amplitude of other ICA components resembling the rest of the functional networks ($p > 0.09$; Fig. 4f and Supplementary Table 8), implicating a specific role of HAR-BRAIN genes in genetic variations of DMN functional activity. Using the normalized DMN amplitude (corrected for the mean amplitude across all networks) as the phenotype of interest showed similar results (HAR genes: $\beta = 0.018$, $p = 0.003$; HAR-BRAIN genes: $\beta = 0.020$, $p = 0.002$; Supplementary Fig. 7).

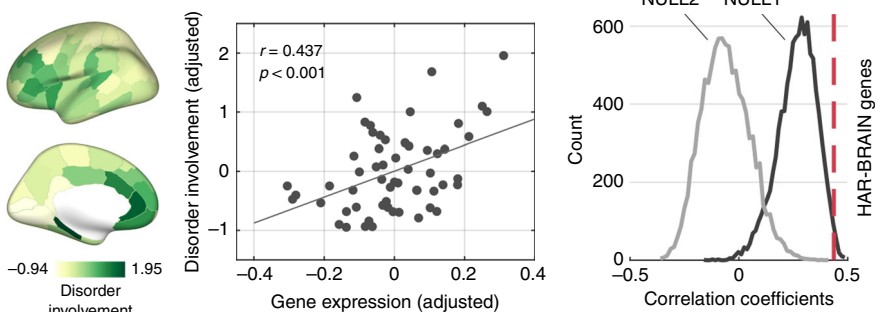

**Fig. 5** Cortical disorder involvement and HAR-BRAIN gene expression. Left panel shows brain maps of cortical involvement across five major psychiatric disorders (e.g., schizophrenia, bipolar disorder, autism spectrum disorder, major depression, and obsessive-compulsive disorder). Middle panel shows correlation between HAR-BRAIN gene expression and disorder involvement (Pearson's $r = 0.437$, $p < 0.001$; corrected for cortical volume), and right panel shows the comparison of the correlation coefficient to null distributions generated by random BRAIN (NULL1; $p = 0.022$) and ECE genes (NULL2; $p < 0.001$, permutation test, 10,000 permutations). Source data provided as Source Data file

Stratified LDSC analysis[29] did not result in findings of significant enrichment of genetic variance of DMN functional activity explained by SNPs overlapping with HAR regions. This is likely related to HARs occupying relatively small regions in the whole genome with a mean length of only 256 base pairs (Supplementary Fig. 8), which limits statistical power to perform such post-hoc analyses. The relatively large s.e. of the SNP heritability might be related to the limited sample size.

**HAR genes, cognitive abilities, and psychiatric disorders**. We then examined the potential role of HAR/HAR-BRAIN and DMN genes in human cognition by cross-referencing these genes with a recent GWAS meta-analysis on intelligence ($h^2_{SNP} = 0.184$ [s.e. = 0.008]) performed on 269,867 individuals[30]. Gene-set analysis[27] revealed both sets of HAR and HAR-BRAIN genes to be significantly associated with individual variations in intelligence (HAR: $\beta = 0.058$, $p = 5.43 \times 10^{-10}$; HAR-BRAIN: $\beta = 0.075$, $p = 1.22 \times 10^{-15}$; Supplementary Table 7). Conditional gene-set analysis[28] including BRAIN genes as a covariate further confirmed a significant association of both HAR and HAR-BRAIN genes with intelligence (HAR: $\beta = 0.056$, $p = 1.60 \times 10^{-9}$; HAR-BRAIN: $\beta = 0.060$, $p = 6.34 \times 10^{-10}$). Intelligence was not found to be specifically associated with the total set of DMN genes ($\beta = 0.012$, $p = 0.061$), but a significant effect was observed for the subset of 37 intersected HAR-DMN genes ($\beta = 0.022$, $p = 0.009$, FDR corrected).

We next linked HAR/HAR-BRAIN and DMN genes to genetic effects on social behavior, which is thought to be more advanced in humans than in other primate species[16]. Summary statistics of the trait "Frequency of friend/family visits" ($h^2_{SNP} = 0.035$ [s.e. = 0.002]) based on a GWAS analysis on 383,941 individuals in the UK Biobank were obtained from the GWAS ATLAS web tool[31] (http://atlas.ctglab.nl, ID 3216). HAR/HAR-BRAIN genes were found to be significantly associated with this trait (HAR: $\beta = 0.037$, $p = 1.24 \times 10^{-7}$; HAR-BRAIN: $\beta = 0.039$, $p = 1.01 \times 10^{-7}$; Supplementary Table 7), with effects unrelated to the set of BRAIN genes (HAR: $\beta = 0.037$, $p = 1.73 \times 10^{-7}$; HAR-BRAIN: $\beta = 0.036$, $p = 2.63 \times 10^{-6}$). Furthermore, DMN genes were also significantly associated with individual variation in sociability ($\beta = 0.012$, $p = 0.024$; HAR-DMN genes: $\beta = 0.014$, $p = 0.022$, FDR corrected).

We also examined the potential association of HAR/HAR-BRAIN and DMN genes with schizophrenia, a disorder hypothesized to relate to human brain evolution[32,33]. We used the summary statistics of a GWAS in 33,426 schizophrenia patients and 54,065 healthy controls ($h^2_{SNP} = 0.187$ [0.008])[34] provided by the Psychiatric Genomics Consortium (http://www.med.unc.edu/pgc/).

We observed HAR/HAR-BRAIN genes to be significantly associated with genetic variants in schizophrenia (HAR: $\beta = 0.019$, $p = 0.013$; HAR-BRAIN: $\beta = 0.043$, $p = 5.06 \times 10^{-7}$; Supplementary Table 7). These results remained significant in the conditional gene-set analysis with BRAIN genes taken as a covariate (HAR: $\beta = 0.017$, $p = 0.023$; HAR-BRAIN: $\beta = 0.028$, $p = 0.001$). DMN genes were not found to be significantly associated with schizophrenia ($\beta = 0.011$, $p = 0.067$; HAR-DMN genes: $\beta = 0.005$, $p = 0.261$). In addition to common variations indicated by GWAS, we further examined the enrichment of HAR/HAR-BRAIN and DMN genes in rare variants of brain disorders using the NPdenovo database (http://www.wzgenomics.cn/NPdenovo/)[35]. Hypergeometric testing showed HAR and HAR-BRAIN genes to be significantly enriched in risk genes of ASD ($p < 0.001$ and $p = 0.005$, separately) and schizophrenia ($p < 0.001$ and $p = 0.008$, separately; FDR corrected). DMN genes also showed significant enrichment in risk genes of ASD ($p = 0.004$), but not schizophrenia ($p = 0.264$; Supplementary Fig. 9).

We also examined a potential association of HAR-BRAIN genes with brain changes related to psychiatric disorders. We used data from voxel-based morphometry (VBM) studies in five psychiatric brain disorders (schizophrenia, bipolar disorder, ASD, major depressive disorder [MDD] and obsessive-compulsive disorder [OCD]) and created a cortical map describing the distribution of cortical volume changes of these five psychiatric disorders (including in total of 260 VBM studies). The spatial pattern of disorder involvement across the cortex was significantly associated with the gene expression pattern of HAR-BRAIN genes ($r(55) = 0.437$, $p < 0.001$, with the cortical volume controlled; Fig. 5; $r(55) = 0.221$, $p = 0.098$ for HAR genes), an effect significantly exceeding the effect obtained by BRAIN genes ($p = 0.022$) and ECE genes ($p < 0.001$, 10,000 permutations). For an out-group analysis, the cortical pattern of HAR-BRAIN gene expression did not correlate to the disease map of five alternative, non-psychiatric disorders (amyotrophic lateral sclerosis, stroke, alcoholism, insomnia, fibromyalgia; $r = 0.141$, $p = 0.302$).

## Discussion

Our combined comparative neuroimaging and genetic findings provide evidence of evolutionary changes in the human genome to have played a central function in the expansion and cortical organization of cognitive functional networks in the human brain, potentially in service of specialization of higher-order cognitive function in human evolution.

Our results show high levels of cortical expansion in regions of both the FPN and DMN in humans. This is compatible with prior observations of cortical expansion between macaque and

human[36], showing large expansion of associative prefrontal, temporal, and parietal areas in the human brain[36,37]. This evolutionary expansion pattern has been suggested to overlap with the pattern of cortical variation in today's human population[38], suggesting larger brains to display relatively larger multi-modal associative areas, variation further linked to inter-subject variation in cognitive abilities[38]. Our observations of high expression of HAR genes in these brain areas now suggest that genes linked to hominization may have had a special role in the process of cortical development of these multimodal association areas.

Our comparative analyses further show that genes associated with human brain evolution (HAR-BRAIN genes) are not equally expressed in all cortical areas but rather are more expressed in areas related to higher-order cognitive processing. HAR genes, representing conserved loci with elevated divergence in humans[17,19], have been argued to function as important neuronal enhancers[39] and to be key players in biological processes of nervous system development and neurogenesis, amongst others (Supplementary Table 9). HAR genes are enriched in human-evolved elements that converge on specific cell types and laminae involved in brain development and cerebral cortical expansion in the primate lineage[40] and are suggested to be particularly expressed in supragranular cortical layers important for forming cortico-cortical connectivity[40]. Our findings of high expression of HAR genes in central cognitive networks, and most pronounced in the DMN, may thus reflect enhanced complexity of cognitive cortical areas and circuits in human brain evolution[41,42].

Our results corroborate previous observations of a strong link between aspects of cellular and macroscale connectivity[43,44]. Association areas show transcriptional profiles enriched for genes specific to the organization of supragranular layers[45], with the spatial layout of functional networks captured by coupled transcription profiles of genes enriched in supragranular layers[46] and genes related to ion channel activity and synaptic function[47]. These observations are further in line with the notion of genes related to the resting-state brain activity of the DMN to display greater expression in neurons[10]. Moreover, our observation of upregulated HAR-BRAIN gene expression in cognitive networks in humans implicates an evolutionarily enhanced complexity of neuronal connectivity in cognitive networks. This might be potentially further related to humans having a longer period of neuronal progenitor expansion compared with chimpanzees and macaques contributing to a differentiated neuronal number and cortical size[48].

Some of the genes found at the intersection of HAR, BRAIN, and DMN genes directly relate to the development of the human central nervous system. For example, CDH8, CDH9, and CDH10 are involved in synaptic adhesion, axon outgrowth, and guidance[49], and play a role in ASD[49]. CBLN1 is important for synapse integrity and synaptic plasticity together with NRXN1 and GRID2[50]. CA10 is believed to be central in the development of the central neural system by coordinating neurexins, which are presynaptic cell-adhesion molecules that bind to diverse postsynaptic ligands and who are linked to several neuropsychiatric disorders[51]. KCNB2 is known to be essential in regulating neurotransmitter release and neuronal excitability[52].

Our findings show that genes highly expressed in the DMN contain genetic variants related to human intellectual ability and sociability. This is compatible with twin studies showing a genetic correlation between IQ and gray matter morphology of DMN regions like the medial frontal cortex and parahippocampal gyrus[53]. The DMN has been argued to be important for human self-projection abilities that include planning the future[54], theory of mind, and navigation[14], of which humans show a higher complexity compared to chimpanzees[55]. This central cognitive system comprises highly connected network hubs like the precuneus and inferior parietal lobule[56], regions involved in multimodal information integration[57], a key aspect of higher-order cognitive brain function. The observation of an association between the spatial pattern of HAR expression and cortical expansion on the one hand, and a significant involvement of HAR genes in genetic variation related to intelligence and social behavior on the other, suggests that the expansion of highly connected hub areas in support of higher-order brain function has been an important driving factor of human brain evolution.

Evolutionary pressure on cognitive networks subserving higher-order brain functions may have been accompanied by an increased risk of brain dysfunction[32,33]. Our comparative findings provide evidence for this hypothesis, with genes important for human brain evolution found to play a role in the development of psychiatric disorders. The pattern of cortical expression of HAR-BRAIN genes shows significant overlap with the pattern of cortical involvement across mental disorders, with particular involvement of lateral and medial prefrontal cortices. These are key 'brain hubs' and components of higher-order networks identified to be generally implicated in the anatomy of a wide range of brain disorders[58]. Our findings further suggest HAR and DMN genes to significantly relate to the genetic architecture for schizophrenia and autism, disorders that are often reported to involve disturbed DMN functional connectivity[59,60]. These findings are consistent with reported genetic associations between the DMN and psychiatric disorders[61] and support the notion of genes related to evolutionary adaptations and brain development to potentially contribute to default-mode network involvement in brain disorders[61].

Several methodological points have to be discussed. We used predefined functional networks to link data from distinct modalities. Network divisions may overlook functional heterogeneity across cortical regions and participation of brain regions in multiple networks, and several other spatial variations of networks are equally viable[62] (see Supplementary Note 5 for alternative definitions of networks, Supplementary Fig. 10–11). Second, the set of HAR genes as used in this study was selected 'as-is' from the study of Doan et al.[19]. HAR-associated genes were labeled as those where HARs are within the introns, within or near (less than 1 kb) 5′ and 3′ UTRs, or are the closest flanking gene that was less than 2.1 mb away (with 70% being less than 500 kb away)[19]. Other mapping approaches can be used to identify and further specify HAR gene sets linked to specific functions. We examined alternative sets of 196 genes mapped from HAR using brain-related Hi-C and eQTL datasets from the PsychENCODE Consortium[63], and a set of 396 genes related to ASD-linked HAR mutations identified using massively parallel reporter assays[19]. These alternative selections and allocations of HAR genes revealed highly consistent findings (data presented in detail in Supplementary Note 1).

Our comparative study shows that recent changes in our genome have played a central role in the expansion and function of higher-order cognitive networks in the human brain. Our findings suggest that expansion of higher-order functional networks and their cognitive properties have potentially been an important locus of change in recent human brain evolution.

## Methods

**Cortical expansion**. In vivo MRI data from 29 adult chimpanzees and 50 adult human subjects were analyzed (see Supplementary Methods for details). Data of chimpanzees were acquired under protocols approved by the YNPRC and the Emory University Institutional Animal Care and Use Committee (IACUC, approval #: YER2001206) (see also Ethics statement). MRI data of humans were obtained from the Human Connectome Project (https://www.humanconnectome.org). T1-weighted scans of chimpanzees and humans were processed using Free-Surfer (v5.3.0; https://surfer.nmr.mgh.harvard.edu/) for tissue classification, cortical ribbon reconstruction, and brain parcellation. Pial surface reconstructions

were used for vertex-to-vertex mapping across chimpanzee and humans and subsequent computation of vertex-wise and region-wise expansion (114-region subdivision of the Desikan-Killiany atlas [DK-114][64,65], 57 per hemisphere; see Supplementary Methods and Supplementary Figs. 12–13). Vertex-wise and region-wise expansion maps are available at https://www.connectomelab.nl/downloads. Validation analysis was performed using the chimpanzee-human BB38 atlas that describes homologous cortical regions between two species[33] (Supplementary Fig. 14).

**Gene expression**. *AHBA*. Cortical gene expression patterns were taken from the transcriptomic data of the Allen Human Brain Atlas (AHBA, http://human.brainmap.org/static/download), including a detailed dataset of microarray gene expression data from brain samples of six human donors (all without a history of neuropsychiatric or neuropathological disorders, demographics tabulated in Supplementary Table 10). Data included expression levels of 20,734 genes represented by 58,692 probes for each cortical region of the left hemisphere[4,66]. Tissue samples were spatially mapped to each of the cortical regions of the FreeSurfer DK-114 atlas[64,65], based on their distance to the nearest voxel within the cortical ribbon of MNI 152 template (and BB38 atlas for validation, see Supplementary Note 4). Samples were normalized to Z scores and averaged across regions (see Supplementary Methods), resulting in a subject × region × gene ($6 \times 57 \times 20{,}734$) data matrix. Normalized gene expression data was averaged across individual datasets to obtain a group level gene expression matrix of the size of $57 \times 20{,}734$.

*PsychENCODE*. Comparative cortical transcription data were obtained from the PsychENCODE database (http://evolution.psychencode.org/)[22], describing batch-corrected, normalized expression levels of 16,463 genes for 11 comparable cortical areas of the human (6 subjects), chimpanzee (5 subjects), and macaque brain (5 subjects, all age and gender controlled and corrected for batch effects[22], see Supplementary Methods, Supplementary Table 11, and ref. [22] for details). Gene expression data were normalized to Z scores across cortical regions within each dataset, resulting in three gene expression matrices (one for each species) of the size of $n \times 11 \times 16{,}463$ ($n = 6/5/5$ for human/chimpanzee/macaque). Data were averaged across individual datasets to obtain a group level gene expression matrix of the size of $11 \times 16{,}463$ for each species.

*HAR genes*. Genes located in human accelerated regions (HARs) of the genome were taken as presented by comparative genome analysis representing genomic loci with accelerated divergence in humans[19]. A total number of 2737 human accelerated regions were identified, representing 2143 HAR-associated genes[19]. One thousand seven hundred and eleven HAR-associated genes were described in the AHBA dataset and used in our analyses, referred to as HAR genes.

*BRAIN genes*. BRAIN genes were selected as the set of genes commonly expressed in human brain tissue using the Genotype-Tissue Expression (GTEx) database (data source: GTEx Analysis Release V6p; https://www.gtexportal.org/). The GTEx portal contains 56,238 gene expression profiles in 53 body sites collected from 7333 postmortem samples in 449 individuals. From these 56,238 genes, a total number of 2823 genes were identified as BRAIN genes showing significantly higher expressions in brain sites than non-brain sites (one-sided $t$-test and an FDR corrected $q < 0.05$ were used). Four hundred and five of these 2823 genes overlapped with the set of HAR genes, referred to as HAR-BRAIN genes.

**DMN genes**. For each of the 20,734 AHBA genes, a two-sided two-sample $t$-test was performed between expression levels of regions of the DMN and regions of the other resting-state networks. Genes showing the top 200 largest $t$-scores (showing $p < 0.004$, uncorrected) were selected and referred to as DMN genes (consistent results were obtained for the set of genes reaching $p < 0.05$, corrected for multiple comparisons and alternatively the set of genes reaching $p < 0.01$ without correction; see Supplementary Note 2 and Supplementary Tables 12–13). Enrichment of HAR genes in top DMN genes was statistically evaluated using hypergeometric test. Gene-set analysis was performed for the set of DMN genes by means of the hypergeometric test implemented in the GENE2FUNC function in FUMA (http://fuma.ctglab.nl)[23] (see Supplementary Methods).

**DMN GWAS**. GWAS was performed on 6899 participants from the UK Biobank (July 2017 release; http://www.ukbiobank.ac.uk; including individuals of European ancestry, relatives excluded). fMRI amplitude of seven ICA-based resting-state networks (described as "NETMAT amplitudes 25" in http://big.stats.ox.ac.uk/; UK Biobank field ID: 25754; for a detailed description, see refs. [25,67] and https://www.fmrib.ox.ac.uk/ukbiobank) were taken as phenotypes of interest. We focused on the phenotype "NETMAT amplitudes 25(01)", describing ICA component #1 resembling the DMN. In addition, ICA component #2, #3, #5, #6, #10, and #14 were examined, respectively, reflecting the VN, VAN, FPN.R, FPN.L, SMN, and LN. GWAS was conducted in PLINK v2.00[68], using an additive linear regression model controlling for covariates of age, sex, twenty European-based ancestry principal components, genotyping array, and total brain volume (derived from the T1 image, linearly transformed to mean zero and variance one). Stringent quality control measures were applied to the summary statistics of the GWAS (see Supplementary Methods and ref. [30] for a detailed description of the used procedures). SNP-based results were mapped and annotated using positional mapping, eQTL mapping, and chromatin interaction mapping as implemented in the SNP2GENE function in

FUMA[23]. MAGMA gene-set analysis was used to examine the association of HAR/HAR-BRAIN/DMN genes with phenotypic variations[27,28].

**Gene-set analysis**. SNP-based summary statistics of three GWAS were obtained, including (1) a recent GWAS meta-analysis on intelligence in 269,867 individuals[30] (https://ctg.cncr.nl/software/summary_statistics); (2) a GWAS of a social interaction related trait, "Frequency of friend/family visits", in 383,941 individuals in the UK Biobank[31] (field ID: 1031; GWAS ATLAS web tool, http://atlas.ctglab.nl/traitDB/3216); (3) a GWAS in 33,426 schizophrenia patients and 54,065 healthy controls[34] as provided by the Psychiatric Genomics Consortium (http://www.med.unc.edu/pgc/). Gene annotation was performed using MAGMA[27], providing gene-based $p$-values and effect sizes that are non-directional and reflect both positive and negative direction given phenotypic variants. Gene-set analysis was performed based on a linear regression model implemented in MAGMA[27] to examine to what extent HAR/HAR-BRAIN and DMN genes are associated with phenotypic variation. Results reaching an FDR corrected $q < 0.05$ were taken as statistically significant (corrected for all 20 tested associations). Conditional gene-set analysis[28] was used to control for the effect of BRAIN genes.

**Cortical involvement in psychiatric disorders**. The BrainMap database was used to assess cortical involvement across five major psychiatric disorders (schizophrenia, bipolar disorder, ASD, MDD, and OCD, including in total of 260 studies) (http://www.brainmap.org). Disease voxel-based morphometry (VBM) data of 260 case-control studies present in BrainMap were extracted using the Sleuth toolbox[69] and meta-analyses were conducted for each disorder using the GingerALE toolbox[70]. Resulting brain maps of activation likelihood estimation (ALE) were registered to the MNI 152 template and regional ALE was computed by averaging ALEs of all voxels within each cortical region of the DK-114 atlas. Regional averaged ALE scores were transformed to Z scores and averaged into a cross-disorder cortical involvement map describing per region the level of involvement across five major psychiatric disorders (see Supplementary Methods).

**Statistical analysis**. Pearson's correlation was used to examine the association of the profile of cortical gene expression with the pattern of chimpanzee-to-human cortical expansion. Two-sided two-sample $t$-test was used to statistically test the difference in evolutionary cortical expansion and mean gene expression of HAR and HAR-BRAIN genes between regions of higher-order cognitive networks (e.g., the DMN, FPN, and VAN) and regions of the SMN/VN. Similar analysis was conducted between each of the functional networks and the rest of the brain. Results reaching an FDR corrected $q < 0.05$ were taken as statistically significant (corrected for eight tests in each analysis). Cohen's $d$ was computed, as the difference between two means divided by a standard deviation, to indicate the effect size. Permutation testing (10,000 permutations) was used to differentiate effects of HAR-BRAIN genes from effects of general BRAIN genes (referred to as NULL1) and genes associated with evolutionarily conserved elements of the human genome (ECE genes, referred to as NULL2). ECE genes were obtained from evolutionarily conserved elements in the human genome with length larger than 200 base pairs as described in ref. [21] and were mapped to genes when they fall inside the genomic location provided by the gene[27], resulting in a set of 9125 genes. For each permutation (for NULL1 and NULL2), 415 genes (the same size as the number of HAR-BRAIN genes) were randomly selected from the pool of 2979 BRAIN genes or 9125 ECE genes, separately, and the same statistics (e.g., Pearson's correlation or $t$-test) were computed for this random set to generate an empirical null-distribution (i.e., noted as the NULL1 distribution for BRAIN genes and NULL2 distribution for ECE genes). The original effects were assigned a two-sided $p$-value by comparing to the null-distributions, according to the proportion ($P$) of random permutations that exceeded the original statistics of HAR-BRAIN genes ($p = P \times 2$ if $P < 0.5$, otherwise $p = (1-P) \times 2$).

**Ethics statement**. Data of chimpanzees were acquired under protocols approved by the YNPRC and the Emory University Institutional Animal Care and Use Committee (IACUC, approval #: YER2001206). No new chimpanzee MRI data was acquired for this study; all chimpanzee MRIs were obtained from a data archive of scans obtained prior to the 2015 implementation of U.S. Fish and Wildlife Service and National Institutes of Health regulations governing research with chimpanzees.

**Reporting summary**. Further information on research design is available in the Nature Research Reporting Summary linked to this article.

## Data availability
Human gene expression data that support the findings of this study are available in the Allen Brain Atlas ("Complete normalized microarray datasets", http://human.brain-map.org). Comparative gene expression data that support the findings of this study are available in the PsychENCODE (Human Brain Evolution) ("Gene expression in RPKM (batch correction by Combat)", http://www.evolution.psychencode.org). The GTEx data that support the findings of this study are available in the GTEx Portal V6p release ("The median RPKM by tissue", https://www.gtexportal.org). The human MRI data (in the part of the cortical expansion) that support the findings of this study are available from the

Human Connectome Project (Q3 release, https://www.humanconnectome.org). The chimpanzee MRI data that support the findings of this study are available as part of the National Chimpanzee Brain Resources (MRI database, https://www.chimpanzeebrain.org). The chimpanzee-human expansion map is available at https://www.connectomelab.nl/downloads. The genotype data used for the GWAS on DMN functional amplitude that support the findings of this study are available in the UK Biobank (application 16406; https://www.ukbiobank.ac.uk). The GWAS summary statistics for "intelligence" that support the findings of this study are available from https://ctg.cncr.nl/software/summary_statistics. The GWAS summary statistics for "Frequency of friend/family visits" that support the findings of this study are available from the GWAS ATLAS webtool (http://atlas.ctg.nl/traitDB/3216). The GWAS summary statistics for "schizophrenia" that support the findings of this study are available from the Psychiatric Genomics Consortium (http://www.med.unc.edu/pgc). The cross-disorder VBM data that support the findings of this study are available in the BrainMap (Sleuth, http://www.brainmap.org). The source data underlying Figs. 2c-d, 3a-f, 3h, and 5, and Supplementary Figs 1, 2, 4, and 14 are provided as Source Data files.

## Code availability

All code is available from the corresponding author upon reasonable request.

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

## Acknowledgements

The work of M.P.v.d.H. was supported by an ALW open (ALWOP.179) and VIDI (452-16-015) grant from the Netherlands Organization for Scientific Research (NWO) and a Fellowship of MQ. Y.W. was supported by the China Scholarship Council (201506040039). P.R.J. was supported by the Sophia Foundation for Scientific Research (SSWO, grant s14-27). L.L. was supported by the National Institute of Mental Health (MH100029). D.P. was supported by The Netherlands Organization for Scientific Research (NWO VICI 453-14-005). Primate work was supported by National Institutes of Health Grants P01AG026423 and National Center for Research Resources P51RR165 (superseded by the Office of Research Infrastructure Programs/OD P51OD11132) to the Yerkes National Primate Research Center, and by the National Chimpanzee Brain Resource, R24NS092988. Human neuroimaging data was kindly provided by the Human Connectome Project, WU-Minn Consortium (Principal Investigators: David Van Essen and Kamil Ugurbil; 1U54MH091657) funded by the 16 NIH Institutes and Centers that support the NIH Blueprint for Neuroscience Research; and by the McDonnell Center for Systems Neuroscience at Washington University. The Genotype-Tissue Expression (GTEx) Project was supported by the Common Fund of the Office of the Director of the National Institutes of Health, and by NCI, NHGRI, NHLBI, NIDA, NIMH, and NINDS. The genetic analyses were carried out on the Genetic Cluster Computer, which is financed by the Netherlands Scientific Organization (NWO: 480-05-003), Vrije Universiteit, Amsterdam, The Netherlands, and the Dutch Brain Foundation, and is hosted by the Dutch National Computing and Networking Services SurfSARA. We would like to thank Mats Nagel (VU Amsterdam) and Ting Qi (Max Planck Institute for Human Cognitive and Brain Sciences) for helpful discussions and suggestions.

## Author contributions

Y.W. and S.C.d.L. performed the analyses. M.P.v.d.H. conceived the idea of this study and supervised analyses. L.L., T.M.P., and J.K.R. collected chimpanzee MRI data. L.H.S. and D.J.A. contributed to chimpanzee data processing. P.R.J., J.E.S., and K.W. prepared genetic data. D.P. supervised the genetic analysis pipeline. Y.W. and M.P.v.d.H. wrote the paper with contributions from all coauthors.

## Competing interests

The authors declare no competing interests.
