## [Peer Review File · Nature Communications]

Reviewers' comments:

Reviewer #1 (Remarks to the Author):

Authors have successfully addressed all concerns I raised. The manuscript was greatly improved in the process of revision, and I believe it is now suitable for publication.

Reviewer #2 - was not available to review, but the following are comments from one of the other reviewers:

[REDACTED]

Authors' discussion point is indeed useful, but it would be still encouraged to add a discussion point about how HARs, which are largely hypothesized to mediate cortical expansion during brain development, may be also related to the postnatal cortical connectivity. A recent paper on HARs (<https://www.ncbi.nlm.nih.gov/pubmed/31160561>) raised a point that HAR genes are associated with cortical connectivity and elaboration, which can be useful for authors.

[REDACTED]

The result authors presented is indeed interesting, but according to the method section that authors provided, it is unclear whether the expression values of human, chimps, and macaque brain samples were adjusted between batches, ages, sex, and other confounding variables. In particular, given that overall expression values of HAR genes differ between human chimp, and macaque in Figure 3h, it is important to make sure that the normalization between different samples has been conducted correctly.

Also, the Y-axis of Figure 3h is unclear in terms of what it means. Assuming that it's referring to Z-scores as authors have stated in the Methods, I would say that the major difference between human and chimps does not come from the absolute differences between Higher-order networks and SMN/VN (delta Z-score in human ~ 0.2, delta Z-score in chimps ~ 0.2), but rather the variance within the samples (individual points from human samples are more centered to the mean, while individual points from chimp are more widely distributed). To show the robustness of this difference, authors may use randomly permuted sets of genes (evolutionary conservation matched or overall expression values matched) do not show this pattern, while HAR genes exclusively show differences between human and chimp/macaque.

[REDACTED]

Reviewer #3 (Remarks to the Author):

I would like to thank the authors for carefully addressing my prior questions and comments in their revisions.

Reviewers' comments:

Reviewer #1 (Remarks to the Author):

Authors have successfully addressed all concerns I raised. The manuscript was greatly improved in the process of revision, and I believe it is now suitable for publication.

We are pleased to hear that the reviewer is satisfied with our responses and modifications. We again thank the reviewer for his/her time assessing our manuscript and constructive suggestions that greatly improved our manuscript.

Reviewer #2 - was not available to review, but the following are comments from one of the other reviewers:

Reviewer #2 mentioned before

[REDACTED]

Authors' discussion point is indeed useful, but it would be still encouraged to add a discussion point about how HARs, which are largely hypothesized to mediate cortical expansion during brain development, may be also related to the postnatal cortical connectivity. A recent paper on HARs (<https://www.ncbi.nlm.nih.gov/pubmed/31160561>) raised a point that HAR genes are associated with cortical connectivity and elaboration, which can be useful for authors.

We thank the reviewer for pointing out this interesting paper recently published in Nature Communications. We added a discussion point in the manuscript:

Discussion, Page 15:

[...] HAR genes are enriched in human-evolved elements that converge on specific cell types and laminae involved in brain development and cerebral cortical expansion in the primate lineage⁴⁰ and are suggested to be particularly expressed in supragranular cortical layers

important for forming cortico-cortical connectivity⁴⁰. Our findings of high expression of HAR genes in central cognitive networks, and most pronounced in the DMN, may thus reflect enhanced complexity of cognitive cortical areas and circuits in human brain evolution^{41, 42}.

Reviewer #2 mentioned before

[REDACTED]

The result authors presented is indeed interesting, but according to the method section that authors provided, it is unclear whether the expression values of human, chimps, and macaque brain samples were adjusted between batches, ages, sex, and other confounding variables. In particular, given that overall expression values of HAR genes differ between human chimp, and macaque in Figure 3h, it is important to make sure that the normalization between different samples has been conducted correctly.

We apologize for not being clear on this. We first obtained the gene expression data of the human, chimpanzee, and macaque directly from the *Science* paper of Sousa et al., (2017). As described in their study, Sousa et al. performed batch effect correction by means of the R package *ComBat*, resulting in cross-group normalized data. We apologize for leaving this out, and we now added this information to the manuscript. Given this normalized gene expression data, we additionally performed *z*-score transformation across brain areas within each individual, as we performed for the ABHA data, as suggested by Arnatkevičiūtė et al., (2019). Regarding the age and sex of the included samples, we now describe the individual sample data as presented by Sousa et al. in Supplementary Table 14. As taken from the publication of Sousa et al., the age of all subjects was matched and correspond to respective young to early middle adulthood. Sex was matched across species.

We added this information in the Methods, Supplementary Methods, and Supplementary Table 14:

Methods, Page 19:

[...] Comparative cortical transcription data were obtained from the PsychENCODE database (<http://evolution.psychencode.org/>)²², describing batch-corrected, normalized expression levels of 16,463 genes for 11 comparable cortical areas of the human (6 subjects), chimpanzee (5 subjects) and macaque brain (5 subjects, all age and gender controlled and corrected for batch effects²², see Supplementary Methods and Supplementary Table 14 and ²² for details). Gene expression data were normalized to Z scores across cortical regions within each dataset, resulting in three gene expression matrices (one for each species) of the size of $n \times 11 \times 16,463$ ($n = 6/5/5$ for human/chimpanzee/macaque). [...]

Supplementary Methods, Page 10:

Transcription data of the human, chimpanzee, and macaque

Cortical transcription data of the human (*Homo sapiens*), chimpanzee (*Pan troglodytes*), and macaque (*Macaca mulatta*) were obtained from the PsychENCODE database (<http://evolution.psychencode.org/>)¹⁵. The PsychENCODE database provides expression levels of 16,463 genes for 16 homologous brain locations (10 cortical, 5 subcortical, 1 limbic) in humans (6 subjects), chimpanzees (5 subjects), and macaques¹⁵ (5 subjects; Supplementary Table 15). The age of specimens of all three species was in their respective young to early middle adulthood, and sex was matched across species. No signs of neuropathological abnormalities were reported in any of the specimens from the three species, as reported by Sousa et al¹⁵. The expression levels of genes were quantified by RPKM (reads per kilobase of exon model per million mapped reads). Batch effects were corrected using R package ComBat¹⁶ to normalize the expression values. We additionally performed Z score transformation across brain areas within each individual to quantify gene expressions within the same scale, as suggested by Arnatkevičiūtė et al. (2019)¹⁷. We used the data from the ten cortical regions out of the total 16 brain regions, which included six regions of the higher-order cognitive networks (e.g., dorsolateral prefrontal, inferior parietal, middle frontal, orbital frontal, superior temporal, and ventral lateral prefrontal cortex) and four regions of the primary networks (e.g., primary auditory, primary visual, primary somatosensory, and primary motor cortex). Normalized gene expression data was averaged across individual brains to obtain a group-level gene expression matrix of size of $11 \times 16,463$ for each of the three species.

Supplementary Table 14:

Supplementary Table 14. Demographics of human, chimpanzee, and macaque specimens included in the PsychENCODE dataset

Number	Species	Sex	Age	Stage	Hemisphere
HSB123	Homo sapiens	Male	37	Adulthood	Right
HSB126	Homo sapiens	Female	30	Adulthood	Right
HSB130	Homo sapiens	Female	21	Adulthood	Left
HSB145	Homo sapiens	Male	36	Adulthood	Right
HSB135	Homo sapiens	Female	40	Adulthood	Right
HSB136	Homo sapiens	Male	23	Adulthood	Right
PTB162	Pan troglodytes	Female	22.5	Adulthood	Left
PTB164	Pan troglodytes	Female	30.8	Adulthood	Right
PTB165	Pan troglodytes	Male	31.2	Adulthood	Right
PTB166	Pan troglodytes	Male	26.4	Adulthood	Right
PTB167	Pan troglodytes	Male	29.8	Adulthood	Right
RMB160	Macaca mulatta	Female	10.7	Adulthood	Left
RMB161	Macaca mulatta	Male	11	Adulthood	Left
RMB196	Macaca mulatta	Female	11	Adulthood	Right
RMB218	Macaca mulatta	Male	7	Adulthood	Left
RMB219	Macaca mulatta	Male	7	Adulthood	Left

Also, the Y-axis of Figure 3h is unclear in terms of what it means. Assuming that it's referring to Z-scores as authors have stated in the Methods, I would say that the major difference between human and chimps does not come from the absolute differences between Higher-order networks and SMN/VN (delta Z-score in human ~ 0.2, delta Z-score in chimps ~ 0.2), but rather the variance within the samples (individual points from human samples are more centered to the mean, while individual points from chimp are more widely distributed). To show the robustness of this difference, authors may use randomly permuted sets of genes (evolutionary conservation matched or overall expression values matched) do not show this pattern, while HAR genes exclusively show differences between human and chimp/maquette.

We thank the reviewer for pointing this out to us; we apologize if our figure was not clear. In the figure, we show the normalized gene expression level (i.e., Z score) of HAR-BRAIN

genes in regions of cognitive networks in comparison to primary regions, in the human, chimpanzee, and macaque. Humans show a larger effect size of the enhanced gene expression in cognitive network regions vs primary networks compared to chimpanzees and macaques. We clarified the legend of Figure 3 to be more clear (see below).

The reviewer makes an excellent suggestion to further include a permutation test in which randomly permuted sets of genes are tested. We greatly like this idea. We followed the reviewer's suggestion and performed additional permutation tests in which we separately compared the Δ effect size between humans and chimpanzees (and between humans and macaques) to null distributions of Δ effect size computed by randomly selecting gene sets from the pool of evolutionarily conserved genes (ECE genes) and BRAIN genes. This analysis revealed that the Δ effect size for HAR-BRAIN genes exceeded null distributions of Δ effect size for ECE genes (human-chimpanzee: $p < 0.001$; human-macaque: $p = 0.026$) and BRAIN genes (human-chimpanzee: $p < 0.001$; human-macaque: $p = 0.090$, n.s.). We believe that the non-significant trend-level effect observed for human-macaque difference for the second null model might be related to the notion that HAR-BRAIN genes (which are argued to be important for evolutionary adaptations of brain function between the human and chimpanzee), do not cover all brain-related genetic differentiations between the human and macaque. Taken together, these findings suggest that HAR-BRAIN genes exclusively showed an enhanced gene expression in cognitive networks in humans as compared to chimpanzees.

The reviewer further suggests to test whether the effect may relate more to the variance within the samples. We performed a *leave-one-out* analysis by computing the difference of mean gene expression (Δ gene expression) between cognitive network regions and primary regions for ten rounds, in each of which one region out of the total of ten regions was excluded from the analysis. The resulting Δ gene expression was compared across species. This analysis showed that the Δ gene expression in humans was larger than that in chimpanzees in 9/10 rounds and that in macaques in all 10 rounds, suggesting a difference in Δ gene expression between humans and chimpanzees/macaques.

Taken together, both analyses further support our argument that HAR-BRAIN gene expression is enhanced in cognitive networks in humans, as compared to chimpanzees and macaques. We thank the reviewer for proposing these excellent suggestions.

We made the following changes in our manuscript:

Figure 3 legend, Page 31:

[...] (h) Normalized expression levels of HAR-BRAIN genes in regions of higher-order networks compared to areas of the SMN/VN in humans ($p < 0.001$), with weaker effects in chimpanzees and macaques. [...]

Results, Page 8-9:

[...] Furthermore, the difference in effect size between humans and chimpanzees was larger than expected based on NULL1 and NULL2 (both $p < 0.001$; human-macaque: NULL1, $p = 0.026$ and NULL 2, $p = 0.090$, only trend-level, not significant [n.s.]; 10,000 permutations; Supplementary Fig. 5).

[...] To reduce the influence of a relatively large variance of expression levels within chimpanzees and macaques (Fig. 3h), we performed a leave-one-out analysis (iteratively leaving out one region at a time) and confirmed a larger mean gene expression difference between cognitive network regions and primary regions in humans in comparison to chimpanzees and macaques (Supplementary Fig. 6). [...].

Supplementary Figure 5 and 6, Page 41-42:

Supplementary Figure 5. HAR-BRAIN gene expression enhancement in cognitive network regions in humans compared to chimpanzees/macacaes. Left: permutation testing shows the Δ effect size of the enhanced HAR-BRAIN gene expression in cognitive network regions between humans and chimpanzees to significantly exceed null distributions of Δ effect size computed by randomly selecting gene sets from the pool of BRAIN genes (NULL1: $p < 0.001$) and evolutionarily conserved genes (ECE genes; NULL2: $p < 0.001$). Right:

permutation testing shows the Δ effect size of the enhanced HAR-BRAIN gene expression in cognitive network regions between humans and macaques to significantly exceed NULL2 ($p = 0.026$), but not NULL1 (two-sided $p = 0.090$). A marginal trend-level effect found for NULL1 might be due to the notion of macaques and humans to be more genetically different as compared to chimpanzees, and the set of HAR-BRAIN genes thus to only partially cover the evolutionarily genetic differentiations between the human and macaque and many genetically differences to remain in the NULL condition.

Supplementary Figure 6. Leave-one-out analysis by computing the difference of mean gene expression (Δ gene expression) between cognitive network regions and primary regions for ten rounds, in each of which one region out of the ten regions was excluded. The resulting Δ gene expression in humans is larger than that in chimpanzees in 9/10 rounds and that in macaques in all 10 rounds.

Previous reviewer 2:

[REDACTED]

We thank the reviewer for mentioning these two relevant articles. Elliott and colleagues performed a great number of GWAS on 3,144 neuroimaging-derived phenotypes, reporting

results (including detailed GWAS summary statistics) in the Brain Imaging Genetics database (Oxford BIG; <http://big.stats.ox.ac.uk>). Looking more closely at this great paper, we indeed noticed that one phenotype (“NETMAT amplitudes (25) 01”) is related to the amplitude of fMRI time series within an ICA-component that resembles a network showing characteristics of the default-mode network (referred to as DMN amplitude). Furthermore, we noticed a higher heritability of the ICA DMN amplitude (“NETMAT amplitudes (25) 01”) ($h^2_{\text{SNP}} = 0.09$ [s.e. = 0.06]) compared to directly computing DMN connectivity (as we included before), suggesting that the additional ICA analysis is reducing/extracting spurious signals in the UK Biobank fMRI data.

Given the notion of a higher SNP heritability of the DMN amplitude (thus being able to pick up more genetic variance related to the DMN as compared to before), we included the GWAS results on this DMN phenotype and replaced our previous results on DMN-FC. We performed a GWAS on 6,899 participants from the UK Biobank with “NETMAT (25) 01 amplitude” as the phenotype of interest, using the pipeline as described in our manuscript. Using this approach, we again observed a significant association between HAR-BRAIN genes and DMN amplitude. This again resulted in the conclusion of HAR-BRAIN genes to be significantly associated with individual variations of the DMN functional activity in the human population.

Using the BIG presented ICA-based phenotypes provides an additional opportunity to examine potential specificity of the association between HAR-BRAIN genes and DMN functional activity, with phenotypes related the other networks also available in the UK Biobank. We thus performed GWAS on six other phenotypes, describing the amplitude of networks such as the VN, SMN, VAN, LN, and FPN. MAGMA gene-set analysis did not show any significant association between HAR-BRAIN genes and these phenotypes, suggesting a specific role of HAR-BRAIN genes in the DMN functional activity. We updated our manuscript with this new DMN phenotype and additional analysis (see below).

In the second paper mentioned by the reviewer, Glahn and colleagues examined as one of the first in the field the heritability of the default mode network using fMRI data of individuals from extended pedigrees. They did not report on any specific genes, but we certainly agree with the reviewer that it is interesting to mention these pioneering findings in the context of our current results.

We made the following changes in our manuscript:

Results, Page 10:

GWAS on DMN functional activity

We then wanted to examine whether HAR/HAR-BRAIN genes played a role in inter-subject variation in default-mode functional activity in today's human population. We performed a GWAS on 6,899 participants from the UK Biobank²⁴ (see Supplementary Methods) with the amplitude of fMRI time series of the independent component analysis (ICA)-based resting-state networks ("NETMAT amplitudes 25"²⁵ as described in <https://www.fmrib.ox.ac.uk/ukbiobank/>) as the phenotypes of interest. Particularly, we focused on the amplitude of the ICA component #1 that resembles the DMN (referred to as DMN amplitude, Fig. 4a). GWAS results for all single-nucleotide polymorphisms (SNP) with minor allele frequency (MAF) > 0.005 were assessed (Fig. 4b). The quantile-quantile plot showed a linkage disequilibrium score regression [LDSC] intercept of 0.999 (standard error [s.e.] = 0.006), with an inflation level of $\lambda_{GC} = 1.005$ and mean χ^2 statistic = 1.012. LDSC-based SNP heritability [h^2_{SNP}] was 0.09 [s.e. = 0.06]. We observed 3 independent ($r^2 < 0.1$) genome-wide significant SNPs ($p < 5 \times 10^{-8}$; Fig. 4b) across 2 genomic loci (Fig. 4c). Furthermore, we annotated 19 significant SNPs ($p < 5 \times 10^{-8}$) with high LD ($r^2 \geq 0.6$) to the 3 independent SNPs using gene-mapping functions in FUMA²³ (see Methods), which resulted in a set of 12 genes (Supplementary Table 8; three genes [*PLCE1*, *NOC3L*, and *SLC35G1*] annotated using brain-related eQTL and Hi-C mappings). Hypergeometric testing²³ showed significant enrichment of the 12 genes in the GWAS catalog²⁶ reported gene-set "plasma clozapine-norclozapine ratio in treatment-resistant schizophrenia" ($p = 1.26 \times 10^{-12}$; Supplementary Table 9). None of the genes overlapped with HAR-BRAIN genes or the top 200 DMN. One gene (*INPP5A*) denoted as a HAR gene.

We further investigated the potential association of HAR/HAR-BRAIN genes with variations in DMN amplitude using MAGMA linear-regression-based gene-set analysis²⁷. We found HAR-BRAIN genes to be significantly associated with the phenotypic variation in DMN amplitude ($\beta = 0.015$, $p = 0.016$, FDR corrected). No significant effect was found for the set of HAR genes ($\beta = 0.011$, $p = 0.051$; Supplementary Table 10) or DMN genes ($\beta = 0.005$, $p = 0.219$). An additional conditional gene-set analysis²⁸ including the set of BRAIN genes as a covariate, further showed a significant association of HAR-BRAIN genes with variations in DMN amplitude ($\beta = 0.014$, $p = 0.022$; HAR genes: $\beta = 0.011$, $p = 0.055$; Fig. 4d). Furthermore, no significant effect was observed when we examined the association

between HAR-BRAIN genes and amplitude of other ICA components resembling the rest of the functional networks ($p > 0.09$; Fig. 4e and Supplementary Table 10), implicating a specific role of HAR-BRAIN genes in genetic variations of DMN functional activity. Using the normalized DMN amplitude (corrected for the mean amplitude across all networks) as the phenotype of interest showed similar results (HAR genes: $\beta = 0.018$, $p = 0.003$; HAR-BRAIN genes: $\beta = 0.020$, $p = 0.002$; Supplementary Fig. 7).

Figure 4, Page 36:

Figure 4. GWAS on DMN activity. (a) DMN component (b) GWAS Manhattan plot showing $-\log_{10}$ -transformed two-tailed p -value for all SNP (y-axis) and base-pair positions along the chromosomes (x-axis). Dotted red line indicates Bonferroni-corrected genome-wide significance ($p < 5 \times 10^{-8}$). (c) Regional plots of the two genomic loci (left, lead SNP: *rs11187838* and right, lead SNP: *rs4593926*). (d) Q-Q plot of SNP-based p -value in (b). Observed $-\log_{10}$ transformed two-tailed p -values of associations with DMN functional activity are plotted against expected null p -values for all SNPs in the GWAS. (e) MAGMA conditional gene-set analysis. $-\log_{10}$ -transformed p -values of the associations between

HAR/HAR-BRAIN genes and DMN functional activity conditional upon BRAIN genes. Dashed line indicates $p = 0.05$. (f) MAGMA gene-set analysis on HAR-BRAIN genes and other “NETMAT amplitude 25” phenotypes representing functional activity in the other functional networks ($-\log_{10}$ -transformed adjusted p -values, FDR corrected). Colors indicate the assignment of functional networks, as in Fig. 2b. Dashed line indicates adjusted $p = 0.05$.

Methods, Page 20:

DMN GWAS

GWAS was performed on 6,899 participants from the UK Biobank (July 2017 release; <http://www.ukbiobank.ac.uk>; including individuals of European ancestry, relatives excluded). fMRI amplitude of seven ICA-based resting-state networks (described as “NETMAT amplitudes 25” in <http://big.stats.ox.ac.uk/>; UK Biobank field ID: 25754; for a detailed description, see ^{25, 67} and <https://www.fmrib.ox.ac.uk/ukbiobank>) were taken as phenotypes of interest. We focused on the phenotype “NETMAT amplitudes 25(01)”, describing ICA component #1 resembling the DMN. Additionally, ICA component #2, #3, #5, #6, #10, and #14 were examined, respectively reflecting the VN, VAN, FPN.R, FPN.L, SMN, and LN. GWAS was conducted in PLINK v2.00⁶⁸, using an additive linear regression model controlling for covariates of age, sex, twenty European-based ancestry principal components, genotyping array, and total brain volume (derived from the T1 image, linearly transformed to mean zero and variance one). Stringent quality control measures were applied to the summary statistics of the GWAS (see Supplementary Methods and ³⁰ for a detailed description of the used procedures).

Reviewer #3 (Remarks to the Author):

I would like to thank the authors for carefully addressing my prior questions and comments in their revisions.

We once again thank the Reviewer for his/her thorough comments and constructive suggestions that were of great help to improve our study.

REVIEWERS' COMMENTS:

Reviewer #1 (Remarks to the Author):

I thank the authors to successfully address all the comments reviewers have raised.